# Investigation of Peptide Toxin Diversity in Ribbon Worms (Nemertea) Using a Transcriptomic Approach

**DOI:** 10.3390/toxins14080542

**Published:** 2022-08-08

**Authors:** Anna E. Vlasenko, Vasiliy G. Kuznetsov, Timur Yu. Magarlamov

**Affiliations:** A.V. Zhirmunsky National Scientific Center of Marine Biology, Far Eastern Branch, Russian Academy of Sciences, 690041 Vladivostok, Russia

**Keywords:** ribbon worms, nemertean toxins, peptide toxins, transcriptomics

## Abstract

Nemertea is a phylum of nonsegmented worms (supraphylum: Spiralia), also known as ribbon worms. The members of this phylum contain various toxins, including peptide toxins. Here, we provide a transcriptomic analysis of peptide toxins in 14 nemertean species, including *Cephalothrix* cf. *simula*, which was sequenced in the current study. The summarized data show that the number of toxin transcripts in the studied nemerteans varied from 12 to 82. The most represented groups of toxins were enzymes and ion channel inhibitors, which, in total, reached a proportion of 72% in some species, and the least represented were pore-forming toxins and neurotoxins, the total proportion of which did not exceed 18%. The study revealed that nemerteans possess a much greater variety of toxins than previously thought and showed that these animals are a promising object for the investigation of venom diversity and evolution, and in the search for new peptide toxins.

## 1. Introduction

Animal poisons and venoms are potential sources of new toxic proteins and peptides, representatives of which have already found applications in many areas as therapeutic agents and physiological tools [1,2,3,4]. A large number of these proteins and peptides have been found in marine organisms, and the majority of studies to date have been focused on toxins from cone snails, sea anemones, fish, jellyfish, sea stars, hydras, sea urchins, sea hares, etc. [5,6]. In recent decades, due to the increasing popularity of next-generation sequencing (NGS) technologies, more results from the genome/transcriptome sequencing of neglected animals have appeared in free databases, which makes it possible to use them as a source of new valuable biological information. This kind of study elucidates the ecological aspects of animals’ adaptation and evolution. As a rule, toxic peptides are part of multicomponent mixtures, i.e., venoms or poisons, which have been studied in many animals such as snakes [7], scorpions [8], spiders [9], sea anemones [10], and jellyfishes [11]. However, the study of less popular animals is no less important and allows us to not only find new bioactive peptides, but also to study the evolution of poison and to better understand adaptation processes by conducting comparative analyses of the compositions of animal toxins [12].

One of these little-studied but promising sources of toxins is the marine worms of the Nemertea phylum consisting of more than 1300 species and subdivided into three phylogenetic groups: Palaeonemertea, Pilidiophora, and Hoplonemertea [13,14]. Nemerteans are a rich source of various toxic compounds [15]. Despite the absence of specialized glands for venom and poison secretion, they contain a multicomponent cocktail of toxins that provides them with protection and allows efficient predation [15]. To date, there have been only four works devoted to the screening of toxins using these approaches in nemerteans. The first toxin screening work was carried out in 2014 by Whelan and colleagues on the transcriptomes of nine nemertean species; the number of putative toxin genes found varied from three to seven in different species [16]. More recent works have demonstrated a greater number of toxin genes in nemerteans. A study of the genome of the heteronemertean species *N. geniculatus* revealed 32 putative toxin genes [17]. In 2020, a proteo-transcriptomic analysis of the hoplonemertean *Am. lactifloreus* was carried out by von Reumont and colleagues, and resulted in the identification of 26 peptides that potentially play a role in prey capture, immobilization, and predigestion [18]. The most recent study (carried out in 2022 by Verdes and colleagues), devoted to the proteo-transcriptomic analysis of *A. valida* venom, demonstrated the presence of 85 putative toxins, classified as potentially predatory, defensive, or having dual functions [19]. Based on the results of Luo and colleagues [17], von Reumont and colleagues [18], and Verdes and colleagues [19], it can be assumed that nemertean transcriptomes may contain a much greater variety of toxin-like transcripts than what was shown by Whelan and colleagues [16].

To date, most of the available studies provided information on toxins in a limited number of ribbon worms species and did not reveal the general principles of toxin content in different nemertean classes. The present research was devoted to revealing the toxin transcripts in 14 species of nemerteans belonging to three classes (Palaeonemertea, Pilidiophora, and Hoplonemertea) using the transcriptomic approach, including *Cephalothrix* cf. *simula*, which was sequenced in the current study. The data obtained allowed the discovery of general trends in the diversity of peptide toxins in nemerteans.

## 2. Results

### 2.1. Transcriptomes Assembly

The transcriptomes of 12 nemertean species (*A. lactifloreus*, *Malacobdella grossa*, *Paranemertes peregrina*, *Carinoma hamanako*, *Cephalothrix hongkongiensis*, *Tubulanus polymorphus*, *Baseodiscus unicolor*, *Hubrechtella ijimai*, *Lineus longissimus*, *Lineus ruber*, *Lineus sanguineus*, and *Riseriellus occultus*) were assembled and annotated using reads, which were downloaded from the SRA. The previously published *N. geniculatus* transcriptome was annotated [17], and the assembly and annotation of the *Ce.* cf. *simula* sequences obtained in the current work were carried out. Table 1 presents data processing statistics. The quality of the final assembly was assessed by BUSCO and ranged from 63.9% (*P. peregrina*) to 93.7% (*L. longissimus*). The largest number of nonredundant annotated unique transcripts was obtained for the *Ce.* cf. *simula* transcriptome, which was assembled using a hybrid approach using reads from two platforms, Illumina and MinION Oxford Nanopore. This resulted in 25,895 open reading frames (ORFs) with unique BLAST hits, with an average number of 11,564 ORFs with unique BLAST hits in the remaining 13 nemertean species. ORFs with unique BLAST hits were obtained by removing redundant contigs with identical annotations according to the method proposed by Ono and colleagues [20]. This approach allowed reduction of transcriptome redundancy by 17.8% (*M. grossa*) to 67.6% (*Ce.* cf. *simula*).

### 2.2. Putative Toxin Transcripts in Transcriptomes

To identify toxins, transcripts were annotated using the Tox-Prot and SWISS-PROT/UniProt (*E*-value 10^−6^) databases. Two of the resulting annotations were compared, and the annotation with the highest *E*-value was considered significant (Appendix A).

A total of 14 nemerteans belonging to three classes (Palaeonemertea, Pilidiophora, and Hoplonemertea) were shown to possess from 12–16 (in *P. peregrina* and *M. grossa*, respectively) to 76–82 (in *N. geniculatus* and *Ce.* cf. *simula*, respectively) toxin transcripts (Figure 1). In palaeonemertean species, on average, 49 transcripts were found, from 32 in Ca. hamanako to 82 in *Ce.* cf. *simula*. The average number of toxins in pilidiophorans was 47, ranging from 33 transcripts in *B. unicolor* and *L. ruber* to 76 transcripts in *N. geniculatus* (Figure 1). The number of toxin transcripts in three annotated hoplonemerteans were 12, 16, and 30 in *P. peregrina*, *M. grossa*, and *A. lactifloreus*, respectively, whereas the average number was about 19 transcripts.

Toxin transcripts were divided into seven groups and assigned to 86 families (Figure 2); up to 14 transcripts belonged to each family. The toxins families were determined using UniProtKB/SWISS-PROT [24] Family & Domains section that provides information about the sequence similarity with other proteins. The most represented group was enzymes, which included 32 families, accounting for 16.7–18.8% (*P. peregrina* and *M. grossa*, respectively) to 36.5–47.8% (*L. sanguineus* and *T. polymorphus*, respectively) of all toxin transcripts. The enzyme group was followed by a group of ion channel inhibitors (10 families), where the toxin transcripts occupied from 15.2–16.3% (*B. unicolor* and *R. occultus*, respectively) to 30.3–39.4% (*L. ruber* and Ca. hamanako, respectively) of all toxin transcripts. These two groups, as well as the group of other toxin candidates, were present in the transcriptomes of all 14 species of ribbon worms. The least represented group was neurotoxins; their toxin transcripts were found in 6 out of 14 nemerteans and only in the Palaeonemertea and Pilidiphora classes. The largest number of neurotoxin families (three families) was found in *L. sanguineus*; and five other species bore one family each. Most of the studied species were shown to possess unique toxin families belonging to all toxin groups, except pore-forming toxins. The largest number of unique toxins, four families, was found in *L. longissimus*, *L. sanguineus,* and *N. geniculatus*; no unique toxins were found in Ca. hamanako and *M. grossa* (Figure 2).

The qualitative compositions of the toxins of the three nemertean classes—Palaeonemertea, Pilidiophora, and Hoplonemertea—had both similarities and differences. Figure 3 demonstrates a Venn diagram of the toxin families’ overlaps between nemertean classes. All the classes had toxin families belonging to all the groups of toxins except neurotoxins, which were only found in palaeonemerteans (one family) and pilidiphorans (four families); toxin families of this group did not intersect between the two classes of nemerteans (Table 2). The largest number of common toxin families between the two classes was found in Palaeonemertea and Pilidiophora species, most of which belonged to the enzyme group. Moreover, most of the unique toxin families of all three nemertean classes were also enzymes (from 37 to 50%).

### 2.3. Nemertean-Specific Peptide Toxin Transcripts

Previous studies have described nemertean-specific peptide toxins [15], including cytotoxins AI–AIV, neurotoxins BI–BIV (*Cerebratulus lacteus*) [25], parbolysin (*Parbolasia corrugatus*) (Berne et al. 2003), and nemertides α-1, α-2, and β (*L. longissimus*) [26]. It was subsequently found that parbolysin has high homology with *C. lacteus* cytotoxin A-III [27]. Two nemertean-specific toxins, cytotoxin A-III and nemertide α-1, were identified in the current study. Cytotoxin A-III was detected in *L. longissimus*, *L. ruber*, *N. geniculatus*, *H. ijimai*, *L. sanguineus*, and *R. occultus* (for the first time in the last three). The nemertide α-1 toxin was identified in *L. ruber*, *L. sanguineus*, and *R. occultus*, as in the study by Jacobsson and colleagues [26]. The failure to identify other nemertean-specific toxins might have been caused by different transcriptome assembly techniques.

In a recent study by von Reumont and colleagues [18], non-nemertean-specific toxin transcripts were also identified, including those that were plancitoxin-like, originally isolated from crown-of-thorns starfish, and those that were actitoxin-like, isolated from sea anemones, named U-nemertotoxin-1 and U-nemertotoxin-2, respectively. According to their study, both toxins were typical for nemerteans representing Palaeonemertea, Pilidiophora, and Hoplonemertea. U-nemertotoxin-1 transcripts were found in the *Am. lactifloreus* and *N. geniculatus* proboscis transcriptomes and the full-body transcriptomes of seven species from all three nemertean classes: *Ce. hongkongiensis*, *Cephalothrix linearis*, *Cerebratulus marginatus*, *T. polymorphus*, *M. grossa*, *P. peregrina*, *L. lacteus*, *L. longissimus*, and *L. ruber*. The transcripts of U-nemertotoxin-2 were found in the proboscis transcriptomes of *Am. lactifloreus* and *N. geniculatus*. In the current study, transcripts corresponding to U-nemertotoxin-1 were found in most nemertean species, except for *P. peregrina* and *H. ijimai*. Transcripts presumably related to U-nemertotoxin-2 were identified in 12 out of 14 nemertean species, with the exceptions being *L. sanguineus* and *H. ijimai*.

### 2.4. Assessment of Distance between Species Based on the Presence/Absence of Toxin Families

Based on the presence/absence of toxin families in 14 nemertean transcriptomes, principal coordinate analysis (PCoA) was performed using Jaccard distance, which expresses the distance between species (pseudo-*F* = 3.099, *p* = 0.0001, PERMANOVA, 10,000 permutations in each test) (Figure 4). The results presented in the figure demonstrate that nemertean species were grouped according to the classes to which they belong.

### 2.5. Preliminary Assessment of Toxin Transcripts Expression

The toxin transcripts abundance was quantified for preliminary assessment of toxin expression in all nemertean species using the Salmon tool (Appendix A) and expressed as a percentage; the transcripts with the highest expression, which accounted for 90% of the total expression of the species, were selected (Table 3). For more accurate toxin gene expression levels, the same specimen-preparing conditions should be provided. In all the studied nemerteans, among the transcripts with the highest expression, there were toxins from all seven groups, except for the hoplonemertean species, in which six groups were identified. For the palaeonemerteans, 90% of the total expression was from 10 to 22 toxin transcripts, and the average expression was the highest in representatives of the neurotoxin group (up to 30.3% in *Ce. hongkongiensis*, from the neurotoxin 20 family) and ion channel inhibitors (up to 33.4% in Ca. hamanako, from the CRISP family) (Table 3). Between 5 and 23 toxin transcripts of pilidiophorans accounted for 90% of the expression, and the highest average expression was shown by representatives of the neurotoxin group (up to 35.3% in *L. longissimus*, from the neurotoxin 02 (plectoxin) family. 02 (plectoxin) subfamily) and proteinase inhibitors (up to 40.2% in *L. sanguineus*, from the venom Kunitz-type family). Among the Hoplonemertea members, between five and eight transcripts from 90% of the most expressed toxins were identified. The most expressed were other toxin candidates (up to 47.8% in *A. lactifloreus*, from the TCTP family) and proteinase inhibitors (up to 56.3% in *P. peregrina*, from the venom Kunitz-type family sea anemone type 2 potassium channel toxin subfamily) (Table 3).

Despite the grouping of nemertean species according to their classes demonstrated by PCoA (Figure 4), the abundance of the toxin transcripts in closely related species varied (Table 4). This was also true for major transcripts, which made up 50% of the total toxin expression. Three species from the *Lineus* genus—*L. longissimus, L. ruber,* and *L. sanguineus*—with three, two, and two major toxins, respectively, showed more differences than similarities; the major toxins comprised the Kunitz-type family (proteinase inhibitor group and ion channel inhibitors) and did not contain pore-forming toxins, enzymes and hemostasis-impairing toxins. Two *Cephalothrix* species possessed three major toxins each and were similar only in terms of the presence of MACPF domain-containing toxins and the absence of hemostasis-impairing toxins (Table 4).

## 3. Discussion

Nemerteans possess various toxins with defensive and offensive functions. These include pyridine toxins (anabaseine, nemertelline, 2,3’-bipyridyl, 3-methyl-2,3’-bipyridyl), tetrodotoxin and its analogues (TTXs), and various peptide toxins. According to current data, pyridine toxins are characteristic of hoplonemerteans [28]. The highest concentration and greatest variety of TTXs are specific to palaeonemerteans [29,30,31,32,33], although trace concentrations have been found in pilidiophorans and hoplonemerteans [32]. Peptide nemertean-specific toxins have been identified in Pilidiophora class representatives [34], and transcripts of non-nemertean-specific toxins have been found in all nemertean classes [16]. Recently, due to the fast development of NGS techniques, the transcriptomic approach has become popular, leading to the complete and efficient identification of peptides and their expression evaluation, which permits the comparison of the mechanisms by which toxins are used in animals. In the current study, we reassembled and annotated the transcriptomes of Whelan and colleagues [16] (*M. grossa*, *P. peregrina*, *T. polymorphus*, *Ce. hongkongiensis*, *L. longissimus*, *L. ruber*, and *Am. lactifloreus* [12]). The transcriptomes of Ca. hamanako, *B. unicolor*, *H. ijimai*, *L. sanguineus*, and *R. occultus* were assembled and annotated from the reads deposited in the SRA (NCBI). The previously published transcriptome of *N. geniculatus* [17] was processed and annotated. In addition, in the present study, the *Ce.* cf. *simula* transcriptome was sequenced, assembled, and annotated for the first time. For all of these transcriptomes, the content of toxin transcripts was evaluated; as a result, a total of 588 toxin transcripts were identified, which were divided into 86 families and assigned to seven groups of toxins according to the annotations from the UniProtKB/Swiss-Prot and Tox-Prot databases (Appendix A). These groups were neurotoxins (5 families), pore-forming toxins (5 families), enzymes (31 families), proteinase inhibitors (10 families), ion channel inhibitors (9 families), hemostasis-impairing toxins (4 families), and other toxin candidates (22 families).

The function of animals’ toxic cocktails is reflected by the composition of toxins and their mechanism of action. The mixtures used to deter predators consist predominantly of compounds that induce an immediate reaction and interfere with fast-acting physiological processes such as nerve transmission. Consequently, many defensive poisons contain toxins that quickly cause paralysis by blocking neuromuscular receptors or acting on pain receptors, causing instant and intense pain [35]. At the same time, venoms used to subdue prey are more diverse in the composition and physiological effects of their toxins [36]. In representatives of all nemertean classes, a mixture of toxins with different activities have been identified—Palaeonemertea and Pilidiophora species contained all seven groups of toxins, and Hoplonemertea species contained six groups; no neurotoxins were found in the latter. Presumably, some of them can play the role of a repellent agent to protect against predators, and others can be used during hunting as immobilizing agents or digestive enzymes. To specify the peptide toxins role for nemerteans, more detailed investigation is necessary to carry out. One of the directions is proteotranscriptomic differential gene expression analyses. To date, there are two articles, devoted to investigation of peptides in mucus, covered the nemertean body and proboscis, a specific weapon organ, demonstrating the characteristics of toxins function based on their expression patterns and proteomic distribution [18,19]. According to Verdes with colleagues, proteins of *Antarctonemertes valida* with insulin-like growth factor-binding domain (identified in the current research in *R. occultus*, *A. lactifloreus*, Ca. hamanako, *Ce. hongkongiensis*, *Ce.* cf. *simula*, *T. polymorphus*, *B. unicolor*, *H. ijimai*, *L. longissimus*, *L. ruber*, *L. sanguineus*, *N. geniculatus*) and galactose-binding domain (identified in the current research in *R. occultus*) (Appendix A) were detected in proboscis and were suggested to be a response for predation; protein with Kunitz domain, identified in all nemerteans studied in the current research except for *Ce. hongkongiensis* (Appendix A)*,* was detected in whole specimen, mucus and proboscis, and presumably had dual functions—predatory and defensive [19]. The research of von Reumont with colleagues provided proteotranscriptomic analysis of *Am. lactifloreus* mucus and found out the secretion of protein toxins, also identified in the current research: lectin (nattectin-like) (identified in *R. occultus*), U-actitoxin (identified in Ca. hamanako, *Ce.* cf. *simula, T. polymorphus*, *L. ruber*, *N. geniculatus*), metalloproteinase M12A (identified in *L. sanguineus*, *N. geniculatus*, *A. lactifloreus),* metalloproteinase M12B (identified in *Ce. hongkongiensis*, *Ce.* cf. *simula*, *T. polymorphus*, *H. ijimai*, *L. longissimus*, *L. sanguineus*, *N. geniculatus*, *M. grossa*), L-amino acid oxidase (identified in *Ce. hongkongiensis*, *Ce.* cf. *simula*, *B. unicolor*, *L. longissimus*, *L. ruber*, *L. sanguineus*, *N. geniculatus*, *R. occultus*), plancitoxin (identified in Ca. hamanako*, Ce. hongkongiensis*, *Ce.* cf. *simula*, *T. polymorphus*, *B. unicolor*, *L. longissimus*, *L. ruber*, *L. sanguineus*, *N. geniculatus*, *R. occultus*, *A. lactifloreus*, *M. grossa*) (Appendix A). Presumably, identified toxins secreted in the mucus could play a defensive role and contribute to predation and the paralysis of prey by facilitating the action of other components of a toxic cocktail [18]. Identified in *An. valida* and *Am. Lactifloreus*, toxic peptides with predatory and/or defensive functions were detected in up to 13 nemertean species, studied here, and the common role of peptides for these species may be assumed.

Based on the presence or absence of toxin families, we analyzed the distance between 14 nemertean species for the first time, demonstrating the grouping of species within classes (Figure 4). The largest number of common toxin families between the two classes was found in Palaeonemertea and Pilidiophora species; the toxins families’ profile of Hoplonemertea species was the most unique (Figure 4, Table 2). The variety of toxin composition of nemertean classes may be caused by their different feeding ecology, including diet preferences and hunting strategies. Thus, it was revealed that palaeonemerteans and pilidiophorans within classes and species individually are characterized by a wide range of potential types of prey: palaeonemerteans prefer nematodes, oligochaetes, polychaetes, and other nemerteans, and pilidiophorans prefer all of the above and additionally bivalves and crustaceans [37,38,39,40,41]. On the contrary, for the Hoplonemertea class, small range of victims is typical, most species prefer one systematic group, mainly crustaceans, as prey, and reject others [38,40,42,43]. Therefore, it could be assumed that the diversity of peptide toxins depends on diversity of potential prey, and since diet of hoplonemerteans is limited to one type of victim, they do not require a great variety of toxic agents. In the case of palaeonemerteans and pilidiophorans, a wide range of potential victims may result in a wide range of toxins, while each type of animal can be affected by a specific toxin. An assumption about the relation between toxin composition and diet preferences was put forth by Verdes and colleagues, due to the revealing of toxin specificity for different nemertean classes [19]. The hunting strategy of hoplonemerteans also increases the discrepancy with other nemertean classes: their proboscis is armed with a stylet that pierces the victim and directly injects the venom cocktail, while Palaeonemertea and Pilidiophora representatives wrap the proboscis around the object without piercing [44,45]. This aspect can also be associated with qualitative composition of toxins and may explain the absence of peptide neurotoxins in the representatives of hoplonemerteans—since the prey capture is accompanied by body piercing, its immobilization is not required.

Despite the similar qualitative composition of toxins in representatives of one class according to PCoA (Figure 4), their quantitative compositions variates greatly. The preliminary expression of toxin transcripts was measured, and it was found that the abundance of the same toxin families differed significantly even in closely related species (Table 4). Thus, in three representatives of *Lineus* genius, two common toxin groups were identified within major toxins between *L. longissimus* and *L. sanguineus* (neurotoxins, proteinase inhibitors), and *L. longissimus* and *L. ruber* (ion channel inhibitors) (Table 4). This may indicate differences in diet preferences of each species: *Lineus* species live in the same area; however, the victims are distinguishing [37,38,40,46]. The abundance of major toxins in another closely related species—*Ce. hongkongiensis* and *Ce.* cf. *simula*—was similar for pore-forming toxins and varied significantly for neurotoxins, enzymes (presented in *Ce. hongkongiensis*) and proteinase inhibitors, ion channel inhibitors (presented in *Ce.* cf. *simula*) (Table 4). The victim types, preferred by this nemertean species, have not been studied, therefore, correlation between diet and toxins composition could not be established. However, the variability in expression levels for the same toxin families can also result from different RNA preparation conditions: *Ce.* cf. *simula* RNA was extracted from the middle of the nemertean body and did not contain proboscis, and *Ce. hongkongiensis* RNA sample was obtained from three individuals and the tissue or body parts were not mentioned, there was no information about proboscis presence (Table 1). The same situation was observed within hoplonemerteans, where all transcriptomes were obtained from dissimilar tissues (Table 1), which correlates with the absence of major toxins overlapping between them (Table 4). Nemerteans toxins are thought to be secreted by glandular cells located in the epidermis of the integument for potential use against predators, and by cells located in the proboscis epidermis to contribute to prey capture [15]. Depending on toxins’ functions, their expression levels in these organs can be different [19]. The provided toxin transcripts abundancy estimation was preliminary and needs to be evaluated using the same tissue type and library preparation conditions.

## 4. Conclusions

Resulting from the transcriptomic analysis, high diversity of toxins and general trends in the distribution of peptide toxins within Nemertea phylum were revealed. The principal coordinate analysis of the distance between 14 nemertean species based on the presence/absence of toxin families in transcriptomes demonstrated that nemertean species were grouped according to the classes to which they belong—Palaeonemertea, Pilidiophora, and Hoplonemertea. The qualitative comparison of the toxin composition of the three nemertean classes showed the toxin families’ overlaps between nemertean classes; the largest number of common toxin families between the two classes was found in Palaeonemertea and Pilidiophora species. The correlation between number of common toxins with evolution distance between the classes is a question for further investigation. Palaeonemertea and Pilidiophora representatives, as the nemerteans with the largest number of toxin transcripts, may be the most promising objects for future studies. The results obtained point to the need for further study of the toxic composition of nemerteans, including proteo-transcriptomic analysis, in order to clarify the spectrum of toxins and study their expression and localization, as well as to search for new, unstudied toxic peptides.

## 5. Materials and Methods

### 5.1. Animal Collection

Specimens of the nemertean species *Ce.* cf. *simula* were collected in October 2019 and August 2020 from rhizoids of *Saccharina* sp. in the Spokoynaya Bay, Peter the Great Gulf, and the Sea of Japan (42.7090° N, 133.1809° E) (Figure 5). The species were kindly identified by Dr. Alexey V. Chernyshev from the Far Eastern Branch of the Russian Academy of Sciences, A.V. Zhirmunsky National Scientific Center of Marine Biology (Vladivostok, Russia). Before RNA isolation, animals were kept in aerated aquaria at 17 °C.

### 5.2. RNA Isolation, Library Preparation, and Sequencing

The total RNA of *Ce.* cf. *simula* middle of body was isolated using TRIzol Reagent (Thermo Fisher Scientific, Waltham, MA, USA), according to the manufacturer’s protocol. RNA concentration and quality were assessed using a BioSpec-nano analyzer (Shimadzu, Kyoto, Japan) at 260/280 nm and 260/230 nm. The length of the fragments was estimated by electrophoresis in 1.2% agarose gel in TAE buffer stained with ethidium bromide with an RNA length marker, i.e., the RiboRuler High Range RNA Ladder (Thermo Fisher Scientific). mRNA was isolated from total RNA using the NEBNext Poly(A) mRNA Magnetic Isolation Module kit (New England Biolabs, Ipswich, MA, USA), followed by double-stranded cDNA synthesis using the Mint2 kit (Eurogen, Moscow, Russia). The result was evaluated by electrophoresis in 1.2% agarose gel in TAE buffer stained with ethidium bromide using the 1 kb DNA ladder DNA length marker (Thermo Fisher Scientific). The double-stranded cDNA was isolated from the reaction mixture with the Bioline ISOLATE II PCR and Gel Kit (Meridian Bioscience Inc., Cincinnati, OH, USA), following the manufacturer’s protocol. The concentration of isolated cDNA was assessed using a Qubit fluorometer (Thermo Fisher Scientific). For sample enrichment with low-represented sequences, cDNA was normalized using the Trimmer-2 kit (Evrogen).

Normalized and double-stranded cDNA sequencing was performed on a MinION Mk1B Oxford Nanopore platform (Oxford Nanopore Technologies, Oxford, UK) using the Direct cDNA Sequencing Kit SQK-DCS109 (Oxford Nanopore Technologies), following the manufacturer’s protocol.

To prepare the cDNA library for the Illumina platform (San Diego, CA, USA), normalized and non-normalized cDNA samples were amplified. The library for Illumina was prepared by the NEBNext Ultra II FS DNA Library Prep Kit for Illumina (New England Biolabs), using the protocol for >100 ng cDNA samples. Sequencing was outsourced to JSC “TsGRM “GENETIKO” (Moscow, Russia).

### 5.3. Transcriptome Assembly and Data Analysis

To assemble the nemertean transcriptomes, we used datasets from the SRA (https://www.ncbi.nlm.nih.gov/sra, accessed on 18 January 2022) that met two requirements: (1) the raw reads contained more than just 3Gbases and (2) after the raw reads were decontaminated, the GC count per read was close to the theoretical distribution, i.e., *Am. lactifloreus* (SRR11906528), *M. grossa* (SRR1507002), *P. peregrina* (SRR1611562), Ca. hamanako (SRR1505094), *Ce. hongkongiensis* (SRR618505), *T. polymorphus* (SRR1611583), *B. unicolor* (SRR1505175), *H. ijimai* (SRR1505100), *L. longissimus* (SRR2682192), *L. ruber* (SRR1324988), *L. sanguineus* (SRR3581110), and *R. occultus* (SRR1505179). The read quality was assessed using the FastQC v0.11.9 software package (https://www.bioinformatics.babraham.ac.uk/projects/fastqc/, accessed on 23 January 2022). Adapter removal and quality filtering (Q ≥ 20) were performed using Trimmomatic v0.39 [47]. De novo transcriptomes were assembled using SPAdes v.3.15.3 [48]. The transcriptome assembly was assessed using BUSCO v5.2.1. [49]. The assembled *N. geniculatus* transcriptome [17] was downloaded from the Marine Genomics Unit (https://marinegenomics.oist.jp/, accessed on 18 January 2022) (Okinawa Institute of Science and Technology).

The *Ce.* cf. *simula* transcriptome assembly was carried out using a pipeline, developed in the current study, based on a hybrid method combining data obtained on the MinION Oxford Nanopore and Illumina platforms. The pipeline included the following steps. First, adapters and chimeric sequences were removed from Oxford Nanopore long reads using Porechop (v. 0.2.4) (https://github.com/rrwick/Porechop, accessed on 23 January 2022). Then, Illumina short reads were corrected using prepared Oxford Nanopore reads by minimap2 (v. 2.24-r1122) (ten cycles) [50] and racon (v. 1.4.13) [51]. Unigenes were obtained using the CAP3 program [52]. Illumina short reads were used separately for de novo transcriptome assembly in SPAdes v.3.15.3. Then, assembled from Illumina short reads, the transcriptome was combined with the corrected Oxford Nanopore long reads using minimap2 (v. 2.24-r1122) and racon (v. 1.4.13).

After assembly, all nemertean transcriptomes were decontaminated using SortMeRNA (v.2.1b) [53] and Seal (from the BBMap v.38.95 package) (https://sourceforge.net/projects/bbmap/, accessed on 23 January 2022) from the noncoding RNA and mRNA of viruses, bacteria, fungi, and humans.

For open reading frame determination (>70 amino acids) and the prediction of protein sequences, TransDecoder (v. 5.5.0) (https://github.com/TransDecoder/TransDecoder, accessed on 24 January 2022) was used. Sequence annotation was carried out through the BLASTP search [54] in the SWISS-PROT database (metazoa, 193,521 sequences) (The UniProt Consortium, 2021), as well as protein domain families (HMMER) [55] in the Pfam database (v. 35.0, 19,632 entries), and open reading frames with the highest homology with known sequences (e ≤ 10–6) were selected. The toxin search (BLASTP) was performed against the Tox-Prot database (7343 sequences) [56]. For further analysis, proteins with identical annotation in the SWISS-PROT/Tox-Prot databases were taken. Transcripts expression in transcripts per million (TPM) was measured using Salmon (v.1.7.0) [57]. Mature proteins sequences were determined using SignalP (v. 6.0) [58].

PCoA was performed using the QIIME 2 software package [59]. Initially, a matrix was formed, in which 86 families of toxins were correlated (0: absence; 1: presence) with the studied nemertean species. Based on this matrix, the Jaccard distance was calculated and the graph of beta diversity was plotted.

## Figures and Tables

**Figure 1 toxins-14-00542-f001:**
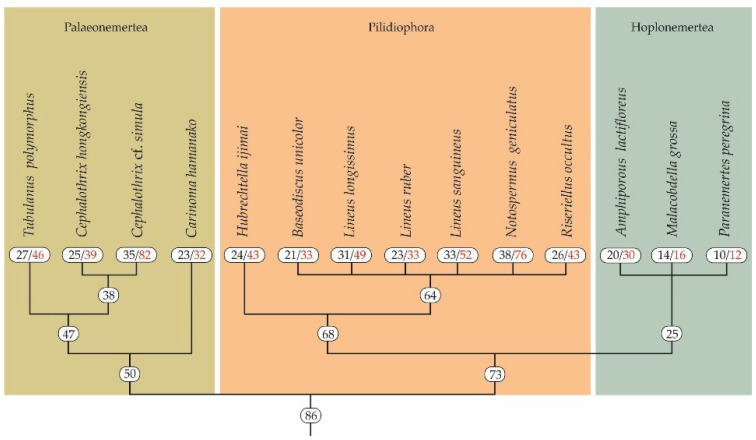
Nemertean phylogenetic tree (phylogeny modified from [21,22,23]). The black figures indicate the number of toxin families, and the red figures indicate the number of toxin transcripts.

**Figure 2 toxins-14-00542-f002:**
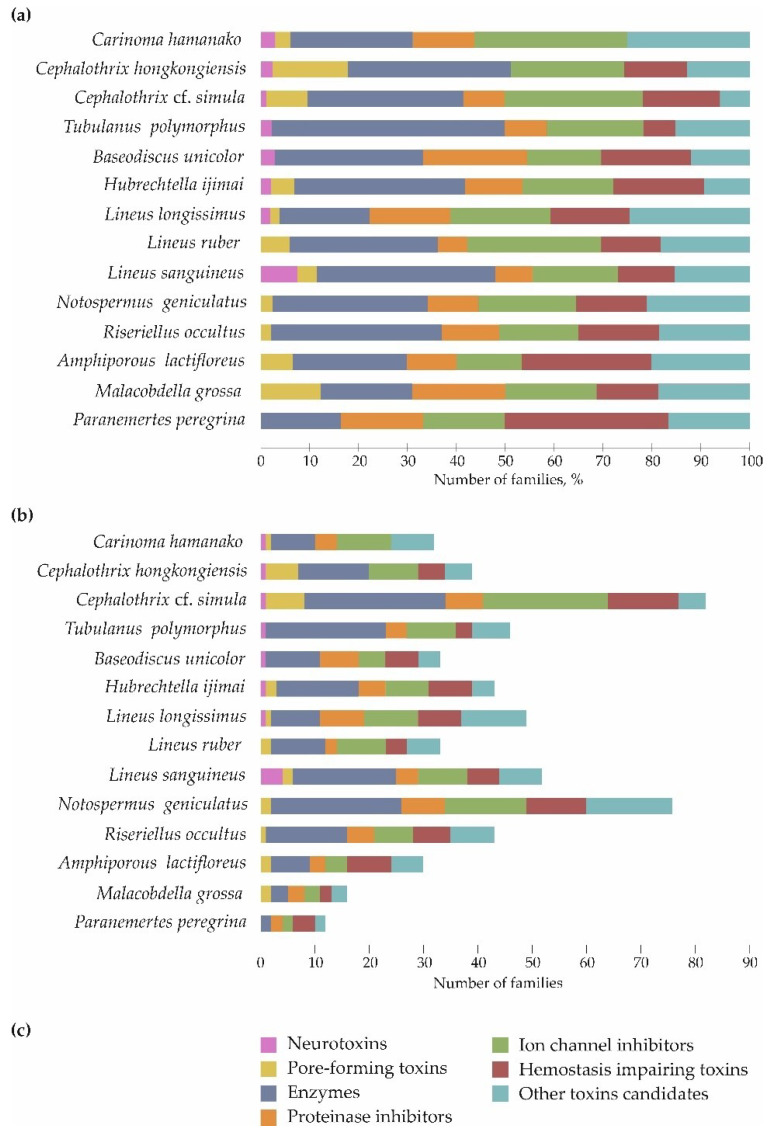
Proportional distributions of toxin families’ transcripts in nemertean transcriptomes. (**a**) Relative distribution, (**b**) absolute numbers, (**c**) the color map for both (**a**,**b**) charts.

**Figure 3 toxins-14-00542-f003:**
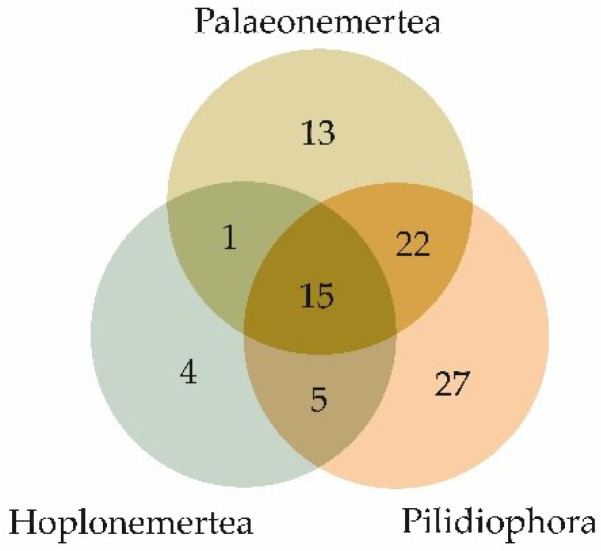
Venn diagram representing the distribution of the numbers of toxin families identified in the three nemertean classes.

**Figure 4 toxins-14-00542-f004:**
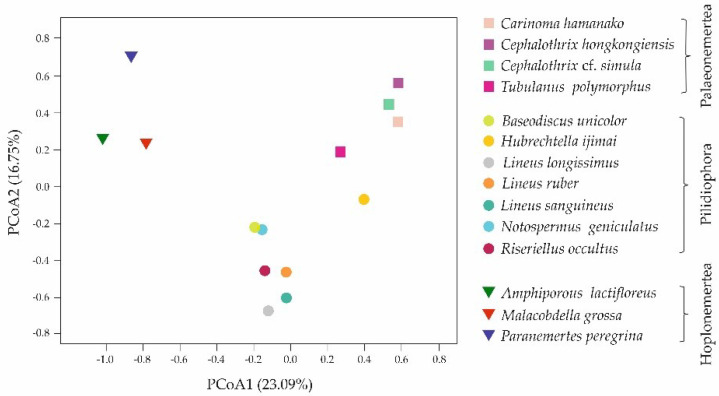
Principal coordinate analysis (PCoA) of the nemertean species based on the presence/absence of the toxin families using Jaccard distance (pseudo−*F* = 3.099, *p* = 0.0001, PERMANOVA, 10,000 permutations in each test).

**Figure 5 toxins-14-00542-f005:**
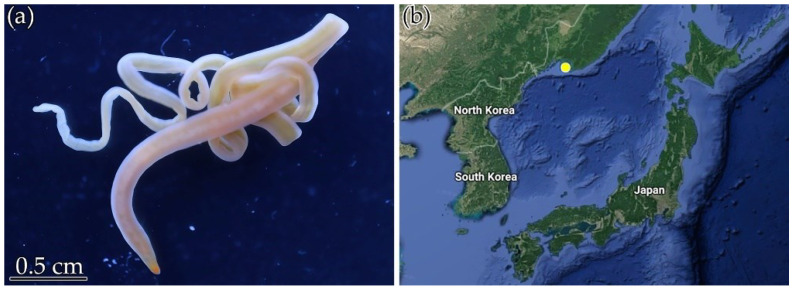
(**a**) A specimen of *Cephalothrix* cf. *simula*, (**b**) location of *Ce.* cf. *simula* collection.

**Table 1 toxins-14-00542-t001:** Assembly statistics of nemertean transcriptomes.

Species	NCBI Run Accession Numbers	Sample Description	Seq Platform	Raw Gigabases	Raw Reads	Size (Gb)	Number of Q20 Reads	Number of Contigs	Number of Decontaminated Contigs	Unigenes	Mean Contig Length	Contig N50	BUSCO, %	Number of ORFs	ORFs with BLAST Hits	ORFs with Unique BLAST Hits
	Palaeonemertea															
*Carinoma* *hamanako*	SRR1505094	Whole individual	Illumina Genome Analyzer II	3.8	52,121,008	2.0	30,851,767	74,088	71,537	67,315	604.7	1206	81.2	25,169	15,209	11,453
*Cephalothrix* *hongkongiensis*	SRR618505	Obtained from three individuals	Illumina Genome Analyzer II	5.2	52,224,518	2.2	23,706,369	101,598	99,702	94,375	451.6	722	78.3	28,675	15,979	11,841
*Cephalothrix* cf. *simula*	SRR18959724 SRR18957873	Middle of body	Illumina NovaSeq6000	16.1	166,114,952	5.2	166,114,952	183,960	125,779	498,451	1132.8	1501	93.1	208,177	79,802	25,895
*Cephalothrix* cf. *simula*	SRR19090815 SRR18968559	Middle of body	MinION Oxford Nanopore	8.6	7,649,450	7.1	–		4,150,437							
*Tubulanus polymorphus*	SRR1611583	Anterior end of body	Illumina HiSeq 2000	3.9	39,262,732	2.6	12,802,492	83,720	81,000	77,268	609.6	1307	82.9	26,069	14,725	11,404
	Pilidiophora															
*Baseodiscus* *unicolor*	SRR1505175	Part of one sample	Illumina Genome Analyzer II	7.1	78,906,444	3.5	78,906,444	272,007	267,551	257,404	637.2	1575	77.6	56,878	18,067	12,335
*Hubrechtella* *ijimai*	SRR1505100	Whole individual	Illumina Genome Analyzer II	4.8	49,505,380	2.5	22,944,699	226,722	221,044	200,648	420.1	790	64.7	36,772	16,640	11,503
*Lineus longissimus*	SRR2682192	Mix of embryonic and juveniles	Illumina HiSeq 2000	43.5	435,278,178	25.4	415,964,152	184,216	172,720	155,885	894.2	3132	93.7	35,740	17,909	12,004
*Lineus ruber*	SRR1324988	Whole body	Illumina HiSeq 2000	5.3	52,196,821	3.3	49,280,205	187,137	184,858	161,352	558.7	1016	80.6	40,754	19,042	12,826
*Lineus sanguineus*	SRR3581110	Whole organism	Illumina HiSeq 2000	11.7	122,617,800	5.8	115,307,676	198,867	195,903	179,731	470.4	919	79.2	46,025	22,599	14,748
*Notospermus geniculatus*		Adult tissues and embryonic stages		Assembly				96,304	94,070	77,565	1286.6	1955	92.2	58,991	24,541	14,520
*Riseriellus occultus*	SRR1505179	Part of one sample	Illumina Genome Analyzer II	3.9	47,787,302	2.0	33,542,144	133,713	131,395	117,212	704.1	1531	66.4	35,025	17,445	12,462
	Hoplonemertea															
*Amphiporous lactifloreus*	SRR11906528	Whole proboscis	Illumina HiSeq 2500	7.0	46,559,252	2.2	39,466,994	55,758	50,950	46,014	1128.8	2464	87.0	23,027	12,214	8928
*Malacobdella grossa*	SRR1611560	1 whole animal	Illumina HiSeq 2000	3.1	30,538,858	2.0	17,132,560	47,700	45,334	42,351	791.0	1905	84.4	15,712	9768	8030
*Paranemertes peregrina*	SRR1611562	Anterior ~1/3 of 1 individual	Illumina HiSeq 2000	5.9	59,441,992	3.6	26,918,767	75,226	69,774	66,797	533.5	1131	63.9	19,744	10,666	8282

–: not applicable.

**Table 2 toxins-14-00542-t002:** Number of toxin families identified in the three nemertean classes.

Nemertean Classes	Neurotoxins	Pore-Forming Toxins	Enzymes	Proteinase Inhibitors	Ion Channel Inhibitors	Hemostasis-Impairing Toxins	Other Toxin Candidates
Common for all							
Palaeonemertea/Pilidiophora/Hoplonemertea		1	3	2	3	3	3
Common for pairs							
Palaeonemertea/Pilidiphora		1	11	1	2		7
Palaeonemertea/Hoplonemertea							1
Hoplonemertea/Pilidiphora			1		1		3
Unique							
Palaeonemertea	1	1	5	1	2	1	2
Pilidiphora	4		10	4	2		7
Hoplonemertea			2	1			1

**Table 3 toxins-14-00542-t003:** Relative abundance of toxin groups’ transcripts identified in three nemertean classes.

Toxin Groups	Hoplonemertea	Palaeonemertea	Pilidiophora
Neurotoxins	NA	+++	+++
Pore-forming toxins	+++	++	+++
Enzymes	+	+	+
Proteinase inhibitors	+++	+	+++
Ion channel inhibitors	+	++	++
Hemostasis-impairing toxins	++	++	+
Other toxins candidates	+++	++	+

NA: not available; +: 0–9.99% of the total sample expression; ++: 10–19.99% of the total sample expression; +++: ≥20% of the total sample expression.

**Table 4 toxins-14-00542-t004:** Relative abundance of toxins that together make up 50% of the total toxin expression (major toxins) in nemerteans.

Protein Family	Tox-Prot Annotation	Expression, %													
		Palaeonemertea				Pilidiophora							Hoplonemertea		
		*Carinoma hamanako*	*Cephalothrix hongkongiensis*	*Cephalothrix* cf. *simula*	*Tubulanus polymorphus*	*Baseodiscus unicolor*	*Hubrechtella ijimai*	*Lineus longissimus*	*Lineus ruber*	*Lineus sanguineus*	*Notosermus geniculatus*	*Riseriellus occultus*	*Amphiporous lactifloreus*	*Malacobdella grossa*	*Paranemertes peregrina*
**Neurotoxins**															
Neurotoxin 20 family.	U3-aranetoxin-Ce1a		30.3												
Neurotoxin 02 (plectoxin) family. 02 (plectoxin) subfamily	Omega-plectoxin-Pt1a							35.3							
Neurotoxin 10 (Hwtx-1) family. 15 (Hntx-3) subfamily	Hainantoxin-III 8									21.9					
Pore-forming toxins															
MACPF domain	Perivitellin-2 67 kDa subunit													33.2	
	DELTA-alicitoxin-Pse2a		17.2	14.7											
Worm cytolysin family	Cytotoxin A-III								16.0		33.8				
Enzymes															
Phospholipase A2 family. Group III subfamily	Phospholipase A2 isozymes PA3A/PA3B/PA5		6.0										21.9		
Arthropod phospholipase D family. Class II subfamily. Class IIb sub-subfamily	Dermonecrotic toxin LiSicTox-alphaV1				16.8	23.9									
	Dermonecrotic toxin LhSicTox-alphaIA1ii						17.9								
Proteinase inhibitors															
Venom Kunitz-type family. Sea Anemone type 2 potassium channel toxin subfamily	PI-actitoxin-Axm2b														56.3
	Actinia tenebrosa protease inhibitors					21.6	18.9								
Venom Kunitz-type family	Kunitz-type serine protease inhibitor conotoxin Cal9.1a					14.5				40.2					
	Kunitz-type serine protease inhibitor conotoxin Cal9.1d							5.0							
	Kunitz-type serine protease inhibitor microlepidin-1										23.6				
	Kunitz-type serine protease inhibitor Bt-KTI											18.7			
Natriuretic peptide family	Snake venom metalloprotease inhibitor 02A10			28.9											
Ion channel inhibitors															
Sea anemone structural class 9a family	Delta-actitoxin-Amc1a													27.8	
CRISP family	CRISP/Allergen/PR-1	30.4													
	Cysteine-rich venom protein Mr30							12.6							
	Cysteine-rich venom protein pseudechetoxin-like											17.9			
	Cysteine-rich venom protein											18.9			
CRISP family. Venom allergen 5-like subfamily	Scoloptoxin SSD976			9.3			8.2								
Venom Kunitz-type family. Sea anemone type 2 potassium channel toxin subfamily	U-actitoxin-Avd3h								35.6						
	U-actitoxin-Avd3l				6.7										
Hemostasis-impairing toxins															
Snaclec family	Snaclec 3				15.9										
Other toxins candidates															
TCTP family	Translationally controlled tumor protein homolog OS=Branchiostoma belcheri												47.8		
	Translationally controlled tumor protein homolog OS=Brassica oleracea	21.3													
Insulin-like growth factor-binding protein-related protein 1	Insulin-like growth factor-binding protein-related protein 1				12.5										

## Data Availability

Sequence files and metadata for all samples used in this study have been deposited in the NCBI Sequence Read Archive repository under BioProject PRJNA832799 (accession numbers SRR18968559, SRR18959724, SRR18957873, SRR19090815), https://www.ncbi.nlm.nih.gov/bioproject/PRJNA832799, accessed on 27 May 2022.

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
