# Peer review of "Investigation of Peptide Toxin Diversity in Ribbon Worms (Nemertea) Using a Transcriptomic Approach"

_toxins, 2022, doi:10.3390/toxins14080542_

Round 1
Reviewer 1 Report
The manuscript entitled "Investigation of Peptide Toxin Diversity in Ribbon Worms (Nemertea) Using a Transcriptomic Approach" describes the transcriptomic analysis of 14 nemertean species to study their diversity of peptide/protein toxins.
The manuscript is well-written, concise, and within the journal's scope. However, although the authors found many toxin transcripts, I think the manuscript is focused on describing the results, but the novelty is unclear. I consider that revealing a "greater number of toxin transcripts than previously thought" does not add much value, as this can be expected when not many transcriptomic studies have been done on nemertean species. For instance, did the authors find a new family of toxins? Do the data shed light on aspects of the evolution of their toxins or their mechanism of envenomation?
On the other hand, I think this work could be the starting point for a more comprehensive molecular study of these toxins if combined with proteomic/peptidomic approaches.
Author Response
We sincerely thank the reviewer for careful reading of our manuscript and positive feedback. Our responses to the reviewer’s comments are given below. All changes in response to the reviewers' comments are marked in green.
“However, although the authors found many toxin transcripts, I think the manuscript is focused on describing the results, but the novelty is unclear. I consider that revealing a "greater number of toxin transcripts than previously thought" does not add much value, as this can be expected when not many transcriptomic studies have been done on nemertean species. For instance, did the authors find a new family of toxins? Do the data shed light on aspects of the evolution of their toxins or their mechanism of envenomation?
On the other hand, I think this work could be the starting point for a more comprehensive molecular study of these toxins if combined with proteomic/peptidomic approaches”.
Response: We thank the reviewer for valuable comment. Our study provides comprehensive assessment of the toxic potential of representatives of all classes of the Nemertea phylum and can form the basis for further studies of peptide toxins. For better understanding, the conclusion section was rewritten, lines 326-341: “Resulting the transcriptomic analysis, high diversity of toxins and general trends in the distribution of peptide toxins within Nemertea phylum were revealed. The principal coordinate analysis of the distance between 14 nemertean species based on the presence/absence of toxin families in transcriptomes demonstrated that nemertean species were grouped according to the classes they belong to — Palaeonemertea, Pilidiophora, and Hoplonemertea. The qualitative comparison of the toxin composition of the three nemertean classes showed the toxin families' overlaps between nemertean classes, the largest number of common toxin families between the two classes was found in Palaeonemertea and Pilidiophora species. The correlation between number of common toxins with evolution distance between the classes is a question for further investigation. Palaeonemertea and Pilidiophora representatives, as the nemerteans with the largest number of toxin transcripts, may be the most promising objects for future studies. The results obtained point to the need for further study of the toxic composition of nemerteans, including proteo-transcriptomic analysis, in order to clarify the spectrum of toxins and study their expression and localization, as well as to search for new, unstudied toxic peptides.”
Reviewer 2 Report
Dear Authors,
I looked at the review "Investigation of Peptide Toxin Diversity in Ribbon Worms (Nemertea) Using a Transcriptomic Approach"
The review is interesting.
However, I have few comments:
1- It will be important to image pictures to show how look like those worms.
2- What is the originality of this review compared to two other reviews indicated in the text: Verdes et al. and Goransson et al. Please clearly indicate in your manuscript the originality of this new review compared to the two already existing.
regards
Author Response
We sincerely thank the reviewer for careful reading of our manuscript and positive feedback. Our responses to the reviewer’s comments are given below. All changes and corrections in response to the reviewers' comments are marked in yellow.
“1- It will be important to image pictures to show how look like those worms.”
Response: We have provided the picture of Cephalothrix cf. simula - only species that is available to us, line 351.
2- What is the originality of this review compared to two other reviews indicated in the text: Verdes et al. and Goransson et al. Please clearly indicate in your manuscript the originality of this new review compared to the two already existing.”
Response: We thank the reviewer for valuable comment, the information was added: lines 65-71: “To date, most of the available studies provide information on toxins in a limited number of ribbon worms species and do not reveal the general principles of toxin content in different nemertean classes. The present research was devoted to revealing the toxin transcripts in 14 species of nemerteans belonging to three classes (Palaeonemertea, Pilidiophora, and Hoplonemertea) using the transcriptomic approach, including Cephalothrix cf. simula, which was sequenced in the current study. The data obtained permit to obtain general trends in the diversity of peptide toxins in nemerteans”.
Reviewer 3 Report
This article aims to present a transcriptomic study of 14 nemertean species and to describe results on transcripts encoding toxins. Transcriptome were assembled after reads retrieving on ncbi or after HiSeq/nanopore sequencing for the Ce cf simula specie. Authors found of 588 toxin transcripts gathering into 86 families and performed a global comparative study of families and predictive functional groups found in the different species.
The result section would gain by being divided into paragraphs with relevant subtitles. Author have to be more specific about how they defined toxin families. What is a toxin family and how they assign a toxin to a family? In this section, author do not provide informations about amino acid precursor sequences and mature sequences and about biochemical properties of mature peptides. These informations lack along the manuscript and could improve it quality. In the paragraph presenting the variations of toxin gene expression levels, authors explained that closely related species present different expression levels for toxin genes. Are author sure that RNAs have been prepared in the same conditions (same tissues for example) for all SRA experiments ? These informations lack in the table 1 and deserve to be discussed.
The discussion has to be rewritten. Line 1 to 28 belong to introduction section. These informations would improve and specify introduction. In the discussion authors discuss very few their results and present more exhaustively what has been found in other venomous species without specific link with their own results.
The word “utilized” has to be change by “used” or “using”
Author Response
We sincerely thank the reviewer for careful reading of our manuscript and valuable comments. Our responses to the reviewer’s comments are given below. All changes and corrections in response to the reviewers' comments are marked in blue.
“The result section would gain by being divided into paragraphs with relevant subtitles”
Response:
We thank the reviewer for valuable comment, the section was divided into paragraphs.
“Author have to be more specific about how they defined toxin families. What is a toxin family and how they assign a toxin to a family?”
Response:
We thank the reviewer for the comment, the information was added, lines 113-116: “The toxins families were determined using UniProtKB/SWISS-PROT Family & Domains section that provides information about the sequence similarity with other proteins.”
“In this section, author do not provide informations about amino acid precursor sequences and mature sequences and about biochemical properties of mature peptides. These informations lack along the manuscript and could improve it quality.”
Response:
We thank the answer, all information concerning nucleotide sequences of nemertean toxins have been provided in Supplementary File 2. Since the aim of our research was to investigate nemertean toxins on transcriptomic level, no information about biochemical properties of mature peptides can be provided.
“In the paragraph presenting the variations of toxin gene expression levels, authors explained that closely related species present different expression levels for toxin genes. Are author sure that RNAs have been prepared in the same conditions (same tissues for example) for all SRA experiments ? These informations lack in the table 1 and deserve to be discussed”
Response:
We agree with the reviewer, the information was added, lines 197-203: “The toxin transcripts abundance was quantified for preliminary assessment of toxin expression in all nemertean species using the Salmon tool (Tables S1 ‒ S14 in Supplementary File 1) and expressed as a percentage; the transcripts with the highest expression, which accounted for 90% of the total expression of the species, were selected (Table 3). For more accurate toxin gene expression levels, the same specimen preparing conditions should be provided”; lines 271-274: “Additionally, it should be noted that transcriptomes sequences of all studied nemertean species could be obtained from RNA prepared under different conditions from different parts of the body, and this may be the reason of the variability in the abundance of the same toxin families.”
“The discussion has to be rewritten. Line 1 to 28 belong to introduction section. These informations would improve and specify introduction. In the discussion authors discuss very few their results and present more exhaustively what has been found in other venomous species without specific link with their own results”
Response:
We agree with the reviewer, this information was rewritten and moved to introduction section.
“The word “utilized” has to be change by “used” or “using”
Response:
The word was changed.
Round 2
Reviewer 1 Report
No further comments from my side. I agree with the content.
Author Response
We sincerely thank the reviewer for careful reading of our manuscript and positive feedback
Reviewer 2 Report
Dear Authors,
Thank you for your corrections.
Regards
Author Response

(The authors gave the same response as above.)

Reviewer 3 Report
Responses to authors are in blue
We sincerely thank the reviewer for careful reading of our manuscript and valuable comments. Our responses to the reviewer’s comments are given below. All changes and corrections in response to the reviewers' comments are marked in blue.
“The result section would gain by being divided into paragraphs with relevant subtitles”
Response:
We thank the reviewer for valuable comment, the section was divided into paragraphs.
OK
“Author have to be more specific about how they defined toxin families. What is a toxin family and how they assign a toxin to a family?”
Response:
We thank the reviewer for the comment, the information was added, lines 113-116: “The toxins families were determined using UniProtKB/SWISS-PROT Family & Domains section that provides information about the sequence similarity with other proteins.”
OK
“In this section, author do not provide informations about amino acid precursor sequences and mature sequences and about biochemical properties of mature peptides. These informations lack along the manuscript and could improve it quality.”
Response:
We thank the answer, all information concerning nucleotide sequences of nemertean toxins have been provided in Supplementary File 2. Since the aim of our research was to investigate nemertean toxins on transcriptomic level, no information about biochemical properties of mature peptides can be provided.
Even if authors performed a transcriptomic study, it is still possible to provide at least in supplementary data translation of cDNA sequences, sequence alignments for each family and prediction of mature sequences, especially since author address the function of these peptides. Please provide all these informations in supplementary data.
“In the paragraph presenting the variations of toxin gene expression levels, authors explained that closely related species present different expression levels for toxin genes. Are author sure that RNAs have been prepared in the same conditions (same tissues for example) for all SRA experiments ? These informations lack in the table 1 and deserve to be discussed”
Response:
We agree with the reviewer, the information was added, lines 197-203: “The toxin transcripts abundance was quantified for preliminary assessment of toxin expression in all nemertean species using the Salmon tool (Tables S1 ‒ S14 in Supplementary File 1) and expressed as a percentage; the transcripts with the highest expression, which accounted for 90% of the total expression of the species, were selected (Table 3). For more accurate toxin gene expression levels, the same specimen preparing conditions should be provided”; lines 271-274: “Additionally, it should be noted that transcriptomes sequences of all studied nemertean species could be obtained from RNA prepared under different conditions from different parts of the body, and this may be the reason of the variability in the abundance of the same toxin families.”
Author mentioned “should be prepared” or “could be obtained”.
These sentences are not enough specific. Author have to check on genbank how RNA samples were obtained, provide these informations in the table 1 and discuss relative abundance of each families with regard of these data...In discussion authors mention that toxin secreting cells have diverse localization, so tissues/stages of development from which total RNA have been prepared is crucial.
“The discussion has to be rewritten. Line 1 to 28 belong to introduction section. These informations would improve and specify introduction. In the discussion authors discuss very few their results and present more exhaustively what has been found in other venomous species without specific link with their own results”
Response:
We agree with the reviewer, this information was rewritten and moved to introduction section.
The discussion remains very disorganized. In the paragraph 248-271, they actually began to describe the diversity of toxin composition in nemerteans without comparison with their own results. They mentioned the putative defence/predatory function and then they talk about relative abundance of toxin transcripts and barely discussed that point.
In the next paragraph, they present toxins production location in general, mention again predatory/defence function, then nermerteans production location and again different class of toxin characterized in different species of nemerteans… They conclude the discussion by presenting the families they identified in their study and discuss the role of proboscis…
So discussion has to be rewritten and results of authors have to be really discussed.
“The word “utilized” has to be change by “used” or “using”
Response:
The word was changed.
OK
Author Response
We sincerely thank the referee for a careful reading of our manuscript and valuable comments. Our responses to the reviewer's comments are below. All changes in the manuscript are marked in green. The discussion section has been completely reorganized and rewritten, but the English language of this section has not yet been corrected by MDPI editorial service; this will be carried out directly before the final step.
“Even if authors performed a transcriptomic study, it is still possible to provide at least in supplementary data translation of cDNA sequences, sequence alignments for each family and prediction of mature sequences, especially since author address the function of these peptides. Please provide all these informations in supplementary data”.
Response:
The protein sequences and alignments for each family and prediction of mature sequences was added (Supplementary File 2. Sequences of nemertean toxins; Supplementary File 3. Alignments of putative toxins protein sequences).
“Author mentioned “should be prepared” or “could be obtained”.
These sentences are not enough specific. Author have to check on genbank how RNA samples were obtained, provide these informations in the table 1 and discuss relative abundance of each families with regard of these data...In discussion authors mention that toxin secreting cells have diverse localization, so tissues/stages of development from which total RNA have been prepared is crucial.”.
Response:
We thank the reviewer for valuable comment. The information was added to the table 1 and discussed in the discussion section, lines 349-361:”However, the variability in expression levels for the same toxin families can also result from different RNA preparation conditions: Ce. cf. simula RNA was extracted from middle of the nemertean body and did not contain proboscis, and Ce. hongkongiensis RNA sample was obtained from three individuals and the tissue or body parts were not mentioned, there were no information about proboscis presence (Table 1). The same situation was observed within hoplonemerteans, where all transcriptomes were obtained from dissimilar tissues (Table 1), which correlates with absence of major toxins overlapping between them (Table 4). Nemerteans toxins are thought to be secreted by glandular cells located in the epidermis of the integument for potential use against predators, and by cells located in the proboscis epidermis to contribute to prey capture [15]. Depending on toxins’ functions, their expression levels in these organs can be different [19]. The provided toxin transcripts abundancy estimation was preliminary and needs to be evaluated using the same tissue type and library preparation conditions”.
“The discussion remains very disorganized. In the paragraph 248-271, they actually began to describe the diversity of toxin composition in nemerteans without comparison with their own results. They mentioned the putative defence/predatory function and then they talk about relative abundance of toxin transcripts and barely discussed that point.
In the next paragraph, they present toxins production location in general, mention again predatory/defence function, then nermerteans production location and again different class of toxin characterized in different species of nemerteans… They conclude the discussion by presenting the families they identified in their study and discuss the role of proboscis…
So discussion has to be rewritten and results of authors have to be really discussed.”.
Response:
The discussion was completely reorganized, all results have been discussed, lines 266-361.